# Exploring Opioid Prescription Patterns and Overdose Rates in South Carolina (2017–2021): Insights into Rising Deaths in High-Risk Areas

**DOI:** 10.3390/healthcare12131268

**Published:** 2024-06-26

**Authors:** Amirreza Sahebi-Fakhrabad, Amir Hossein Sadeghi, Eda Kemahlioglu-Ziya, Robert Handfield

**Affiliations:** 1Department of Industrial and Systems Engineering, North Carolina State University, Raleigh, NC 27606, USA; asahebi@ncsu.edu (A.S.-F.); asadegh3@ncsu.edu (A.H.S.); 2Department of Business Management, Poole College of Management, North Carolina State University, Raleigh, NC 27695, USA; ekemahl@ncsu.edu

**Keywords:** drug supply chain, opioid crisis, opioid policy analysis, prescription drug monitoring program

## Abstract

With opioid overdose rates on the rise, we aimed to develop a county-level risk stratification that specifically focused on access to medications for opioid use disorder (MOUDs) and high overdose rates. We examined over 15 million records from the South Carolina Prescription Tracking System (SCRIPTS) across 46 counties. Additionally, we incorporated data from opioid treatment programs, healthcare professionals prescribing naltrexone, clinicians with buprenorphine waivers, and county-level overdose fatality statistics. To assess the risk of opioid misuse, we classified counties into high-risk and low-risk categories based on their prescription rates, overdose fatalities, and treatment service availability. Statistical methods employed included the two-sample *t*-test and linear regression. The *t*-test assessed the differences in per capita prescription rates between high-risk and low-risk counties. Linear regression was used to analyze the trends over time. Our study showed that between 2017 and 2021, opioid prescriptions decreased from 64,223 to 41,214 per 100,000 residents, while fentanyl-related overdose deaths increased by 312%. High-risk counties had significantly higher rates of fentanyl prescriptions and relied more on out-of-state doctors. They also exhibited higher instances of doctor shopping and had fewer medical doctors per capita, with limited access to MOUDs. To effectively combat the opioid crisis, we advocate for improved local healthcare infrastructure, broader treatment access, stricter management of out-of-state prescriptions, and vigilant tracking of prescription patterns. Tailored local strategies are essential for mitigating the opioid epidemic in these communities.

## 1. Introduction

The opioid crisis refers to the extensive misuse and addiction to opioids, which covers both prescription medications and illegal substances synthesized or derived from the opium poppy plant [1]. It consists of an escalation in overdose deaths, first driven by natural and semi-synthetic opioids, like oxycodone, until 2010, followed by a spike in heroin deaths post-2010, and more recently dominated by illicit synthetic opioids since 2013, with stimulants now also playing a significant role [2,3,4]. Opioid use disorder (OUD) is a chronic condition with persistent opioid consumption and detrimental consequences to the user [5]. Over 2.1 million individuals in the United States are affected by OUD, with an annual death toll of approximately 80,000 linked to opioid misuse [5,6].

In response to this crisis, several states have enacted regulations, including Prescription Drug Monitoring Programs (PDMPs), opioid prescribing guidelines, and prescription limit laws [7,8], aiming to minimize opioid analgesic exposure and prevent the onset of OUD. However, the widespread illicit supplies and varying results from these interventions have hindered their ability to consistently decrease opioid overdoses [9]. In tandem with these measures, numerous states have also implemented regulations to enhance access to overdose prevention strategies. These efforts include increasing the availability and utilization of naloxone, which is a substance used to reverse opioid overdoses [10], as well as expanding the accessibility of medications for opioid use disorder (MOUDs) [11].

There are three main MOUDs approved by the FDA: methadone, buprenorphine, and extended-release naltrexone. Buprenorphine helps to reduce drug use and offers protection from overdoses. It is popular because it requires less monitoring and can be given by qualified primary care doctors. Methadone helps to lessen cravings and withdrawal from heroin, but due to its associated risks, it is only dispensed through certified Opioid Treatment Programs (OTPs) by the Substance Abuse and Mental Health Services Administration (SAMHSA). Naltrexone aids in preventing relapses after detox. It is safe with no overdose risk, has effects lasting up to 30 days with just one dose, and is available through certain primary care providers [12,13,14,15].

MOUDs are vital in preventing overdoses and related care complications, but issues like diagnosis, treatment initiation, and patient commitment can limit their impact. This emphasizes the importance of broader adoption and better training for healthcare providers in OUD care [16,17]. Over the past decade, efforts have been made to expand MOUD availability [18]. As a result, 42% of the U.S. population could access all three types of treatments within a 10-mile radius by the end of 2020 [19]. Yet, despite these advances, gaps in accessibility persist. These gaps are notably pronounced in various geographic locations, including both state and county levels, with rural areas facing particular challenges [20]. To address these disparities, numerous studies explored MOUD availability [21,22,23], their effects on overdose rates [24], and the correlation with community-level socio-demographic factors [25].

Building on this, a recent study correlated county-level MOUD availability with overdose rates, creating a straightforward classification system that categorizes counties into high- and low-risk groups. High-risk counties are those with a lower level of access to MOUDs than the national average, coupled with higher-than-average overdose rates [26]. The authors went on to compare the socio-economic and demographic profiles of these county groups, which are key factors in determining the opioid demand [27]. While this approach provides valuable insights into the demand side of the opioid crisis through its focus on characteristics, there is a noticeable gap in research regarding the legal supply side, as represented by opioid prescription records. To our knowledge, no study has yet explored legal prescription data in relation to MOUD access and overdose rates.

Our study aimed to fill this gap by examining the legal supply side of the opioid crisis using county-level data for South Carolina from 2017 to 2021. The increasing prominence of synthetic opioids, like fentanyl, in overdose deaths, especially during the fourth wave of the opioid crisis, motivated our first hypothesis: that these potent synthetic opioids are more frequently prescribed in high-risk counties [3,28]. Further, based on our prior studies, we observed a trend where patients increasingly seek prescriptions from out-of-state doctors, pharmacies, and retail entities following the implementation of local opioid policies [29,30]. We suspect that this behavior is especially prevalent among patients who are likely to have OUD and, therefore, may be more common in high-risk counties. Confirmation of these patterns could result in significant policy implications, not just on MOUD access but also on the regulation of the opioid supply.

## 2. Methodology

### 2.1. Overview

We started by examining the state-level data to gain a broader perspective on the opioid prescription rates and overdose occurrences in South Carolina. This statewide data provided the initial context for the more detailed county-level analysis that ensued. Following this initial step, we utilized the classification method outlined by Haffajee et al. [26] to categorize counties into high risk and low risk based on the availability of MOUDs and the rates of overdose deaths. Subsequently, our study adopted two primary analytical approaches: one that did not consider time and another that explored changes over time. The non-temporal analysis employed a two-way *t*-test to compare the prescription patterns between the high-risk and low-risk counties, focusing on the type of opioids prescribed, the locations of prescribing entities, and the types of dispensing entities. Following this, the temporal analysis utilized linear regression models to understand how these prescription patterns evolved over time, particularly in high-risk counties.

### 2.2. Overdose Deaths and Opioid Treatment Facilities’ Data

This study leveraged a dataset aggregated from the South Carolina Department of Health and Environmental Control (SCDHEC), the National Vital Statistics System, and the CDC WONDER database to examine the instances of fatal drug overdoses between 2017 and 2021 [6,31,32,33,34,35]. In compiling this dataset, deaths that resulted from drug overdoses were categorized by the SCDHEC using ICD-10 codes X40–X44, X60–X64, X85, and Y10–Y14, with a principal emphasis placed on fatalities related to opioids, among other substances. To categorize specific drug types, the SCDHEC employs distinct ICD-10 codes. “Prescription Drugs” are identified through codes T36.0–T39.9, T40.2–T40.4, T41.0–T43.5, and T43.7–T50.8, while “Opioids” are differentiated using codes T40.0–T40.6. Additionally, natural and semi-synthetic opioids, such as morphine, codeine, oxycodone, hydrocodone, hydromorphone, and oxymorphone, are distinct from fentanyl, which is a prevalent synthetic opioid [36]. To accurately compare the overdose deaths across counties, it is crucial to normalize these deaths by population. Without normalization, comparing raw numbers would lack meaningful context. By using the drug overdose death rate, we ensured that the data were adjusted for each county’s population, providing more relevant and insightful comparisons.

Furthermore, in this study, we analyzed three specific medications for OUDs: methadone, buprenorphine, and naltrexone. We chose to analyze methadone, buprenorphine, and naltrexone for their roles in treating OUDs due to their well-established efficacy and safety profiles. Methadone and buprenorphine help to mitigate withdrawal symptoms and cravings through their agonist properties, whereas naltrexone acts as an antagonist to block opioid effects, aiding in relapse prevention. The selection of these medications is supported by their extensive use in clinical practice and their inclusion in treatment guidelines [37,38]. We extracted data regarding methadone by examining the availability of OTPs as listed on the SAMHSA website [39]. Although it is important to acknowledge that OTPs may dispense additional medications, methadone is singularly dispensed through OTPs that have received certification from SAMHSA. The presence of buprenorphine providers was likewise determined via the SAMHSA website [40]. In contrast, the availability of naltrexone was assessed through the prevalence of VIVITROL, which is a brand that signifies an extended-release formulation of naltrexone, information about which is publicly provided by Alkermes Inc. (Dublin, Ireland), the manufacturer of the drug [41]. As of the end of 2021, our data indicate the existence of 24 fully certified OTPs, 647 buprenorphine providers, and 41 naltrexone providers in the state of South Carolina.

### 2.3. Risk Status Classification of Counties

To formulate a risk profile for each county, we adopted the methodology suggested by Haffajee et al. [26]. We first computed the per capita average of each medication’s providers within the state and then labeled each county as “low” or “high” risk based on a comparison with the state average. To be more precise, we had 0.36 SAMHSA-approved OTPs per 100,000 residents in South Carolina, 0.6 naltrexone providers per 100,000 people, and 9.32 per capita buprenorphine providers. Following this phase, we categorized counties as either high or low risk concerning the availability of each treatment. The ultimate MOUD status was determined by majority rule; that is, if more than one treatment fell into the low category, the overall MOUD status was deemed low, and vice versa. This diverged from the approach suggested by Haffajee et al., who combined all three metrics and compared them with the state average. We argue that our methodology is more equitable, as typically, OTPs or naltrexone providers are less numerous, and their scale is vastly different from buprenorphine providers. Thus, lumping all metrics together seems to reflect the number of buprenorphine providers more than the other two treatments.

Furthermore, in order to determine the risk level for each county, we combined the per capita overdose death rates with the availability of MOUDs, creating a four-category risk status index. The state’s yearly average overdose rate per 100,000 residents stood at 19.02. Consequently, any overdose rate lower than this figure was categorized as low, while rates exceeding it were considered high. This comprised (1) counties with a high availability of MOUD providers but a low overdose rate; (2) counties with a high availability of MOUD providers but a high overdose rate; (3) counties with a low availability of MOUD providers but a low overdose rate; and finally, (4) counties with a low availability of MOUD providers but a high overdose rate. These four categories are illustrated in Figure 1. Evidently, the final category is the most concerning and warrants comprehensive comparison with the other groups. As a result, we further designated the final category as “High Risk” and the remaining three as “Low Risk” counties. Subsequently, we aimed to compare the prescription patterns between these categorized counties. Additional details pertaining to each treatment and county can be found in the Appendix A.

### 2.4. Opioid Prescription Data

This research used data from the South Carolina Prescription Tracking System known as SCRIPTS. This large database has about 43 million prescription records from 2014 to 2022. The focus was on the years 2017 to 2021 so that it matched up with the overdose data that was also available for these years. The SCRIPTS database encompasses multiple dimensions, such as prescription attributes (e.g., drug name, volume, days supply, and refill status); drug characteristics (e.g., National Drug Code); and details on the prescriber, dispenser, and patient.

Three main areas were looked at in this study: (1) type of drug, (2) prescriber’s geographical location, and (3) type of dispensing institution.

#### 2.4.1. Type of Drug

The drugs of interest in this study were opioids, particularly fentanyl, oxycodone, and hydrocodone. These substances constituted roughly 77% of the opioid prescriptions in our dataset. Our focus on these specific opioids was twofold: first, oxycodone and hydrocodone are predominant in opioid prescriptions, rendering other types less relevant for this investigation. Second, emerging evidence indicates that fentanyl plays a critical role in overdose incidents, which is corroborated by the available overdose data [16]. It is important to note that stimulants, which have been implicated in overdose rates in recent literature, are not included in our dataset [3].

#### 2.4.2. Prescriber Location

Our emphasis lay on comparing in-state and out-of-state prescribers. Prior research suggests that out-of-state prescribers are more strongly correlated with suspicious prescribing patterns [29,30].

#### 2.4.3. Type of Dispenser

The predominant sources for prescriptions are retail and chain pharmacies. Hence, our analysis focused on these two dispensing categories.

To clarify, the difference between retail and chain pharmacies was as follows:Retail pharmacies: These are typically independent pharmacies owned by individuals or small groups. They often serve local communities and may provide more personalized services tailored to their clientele. These pharmacies are stand-alone, meaning they are not part of a larger chain of stores.Chain pharmacies: These are part of a large network of stores under a single brand or corporate entity. Examples include CVS, Walgreens, and Rite Aid. Chain pharmacies can be found nationwide, offering standardized services and products.

#### 2.4.4. Doctor Shopping

“Doctor Shopping” refers to the practice of seeking treatment from multiple healthcare providers, either within a single illness episode or with the intention of obtaining prescription medications unlawfully [42,43]. In our study, “Doctor Shopping” referred to patients who sought controlled substance prescriptions from several providers, often not for genuine medical reasons [44,45,46]. Engaging in this behavior exposes individuals to risks such as OUD and deadly overdoses, and they often simultaneously face mental health challenges, alcohol struggles, and come from lower economic backgrounds [47,48,49,50]. Given the severe consequences, both health-wise and legally, identifying “doctor shoppers” is vital for healthcare and law enforcement. In our research, while the SCRIPT data does not explicitly label “doctor shopping”, we extracted instances of this behavior according to the widely accepted definition where patients obtained medication from at least four different prescribers and four dispensers within a year. This approach was chosen based on its frequent use, as pointed out by [51].

### 2.5. Statistical Analysis

To identify differences between high-risk and low-risk counties, we utilized the two-sample *t*-test and linear regression, with each tailored to the specific data and research question. The details are as follows:Two-sample *t*-test: This test compared the mean per capita prescription rates from Table 3. It was used for continuous, normally distributed variables to determine whether differences between the high-risk and low-risk group means were statistically significant or merely due to random chance. In essence, it evaluated whether elements like fentanyl prescriptions or out-of-state prescriptions differed meaningfully between the high-risk and low-risk counties. A resulting *p*-value below 0.05 indicates a statistically significant difference in means.Linear regression for the trend analysis: This method assessed how a dependent variable, like the difference in per capita prescription, varied with an independent variable, often time, spanning 2017 to 2021. This is demonstrated in Figures 3–6. The regression line’s slope signifies the rate of change: a positive slope indicates an increase, while a negative one indicates a decrease. A *p*-value less than 0.05 confirmed the trend’s statistical significance.

By using these two statistical methods in combination, we were able to conduct a nuanced analysis that looked at both the current state and trends over time in high-risk and low-risk counties.

## 3. Results

### 3.1. Opioid Trends in South Carolina: Setting the Stage for County Prescription Comparisons

Before we dive into the county-specific details, we focus on the high-level metrics of opioid use in South Carolina. From 2017 to 2021, the rate of opioid prescriptions per 100,000 residents dropped from 64,223 to 41,214. This happened even though the state issued more than 13 million opioid prescriptions in total. As shown in Table 1, prescriptions for all types of opioids were on a downward trend. In contrast, overdose deaths, particularly from fentanyl, were on the rise, as Table 2 shows. Deaths from natural and semi-synthetic opioids, like methadone, hydrocodone, and oxycodone, also increased but not as much. These contrasting trends made it crucial to examine what happened at the county level.

As depicted in Figure 1, nine counties exhibited below-average MOUD availability coupled with above-average overdose rates, positioning them in a high-risk category, whereas 37 counties did not fall into this category. Figure 2 visualizes the annual overdose death rates, aligning with the broader statewide trends. Notably, overdoses linked to fentanyl saw an increase, predominantly in the high-risk counties, with a marked rise in 2020 and 2021. In contrast, overdoses that resulted from natural and semi-synthetic opioids remained relatively stable, and their prevalence reduced over time, persisting in both the high-risk and low-risk counties.

These patterns suggest that the counties identified as high-risk were grappling with distinct challenges, particularly those related to fentanyl overdoses. Subsequently, we aimed to investigate whether insights could be derived from the analysis of legal opioid prescriptions within these counties to further comprehend these variances.

It is important to mention that the overdose data include both legal and illegal drugs. Since we only have access to data on legally prescribed opioids, our research only considered the correlation between legal prescriptions and county risk status.

### 3.2. Comparing Prescription Patterns in High- and Low-Risk Counties

Our analysis of the prescription data revealed key differences between the high-risk and low-risk counties when it came to opioid use, as shown in Table 3. Even though patients in both the high-risk and low-risk counties had about the same number of opioid prescriptions per person, there was a key difference: patients in the high-risk counties had 17% more fentanyl prescriptions per person. The statistical tests confirmed that this was a significant difference. On the other hand, the prescriptions for natural and semi-synthetic opioids were about the same in both types of counties.

In addition, in the high-risk counties, patients were 75% more likely to obtain prescriptions from out-of-state doctors, which was statistically significant too. When it came to prescriptions from doctors within the state, the rates were about the same for both the high-risk and low-risk counties. When it came to picking a pharmacy, there was not a big difference. Patients in high-risk areas used retail pharmacies about 12% more, but this was not a statistically significant difference. The use of chain pharmacies was about the same in both the high-risk and low-risk counties.

Next, we considered the time and the interaction between each pair of characteristics listed in Table 3. To come up with a metric that applied to both the high-risk and low-risk counties, we calculated what we called the “Differential Percentage”. This metric captured the difference in average per capita prescriptions between the high-risk and low-risk areas, normalized by the average per capita prescription in the low-risk areas and multiplied by 100.
$$\begin{array}{cc} \mathrm{Differential}\;\mathrm{Percentage} & =\frac{\mathrm{Average}\;\mathrm{in}\;\mathrm{high}\text{-}\mathrm{risk}-\mathrm{Average}\;\mathrm{in}\;\mathrm{low}\text{-}\mathrm{risk}}{\mathrm{Average}\;\mathrm{in}\;\mathrm{low}\text{-}\mathrm{risk}}\times 100 \end{array}$$

#### 3.2.1. Opioid Type and Prescriber Location

In Figure 3, each point represents the differential percentage, as defined in the previous section, for fentanyl prescriptions obtained from in-state and out-of-state doctors. In each subplot, we also provide two indicators: the mean and the slope. The mean reflects the average of the differential percentage, with its significance determined through a *t*-test. The slope indicates the coefficient in a linear regression, offering insight into whether the differential percentage was increasing or decreasing over time. This figure illustrates a higher prevalence of opioid prescriptions, particularly fentanyl, in the high-risk counties from out-of-state prescribers, aligning with insights from Table 3. This figure denotes a rising trend for fentanyl in the high-risk regions over time. In contrast, natural and semi-synthetic opioids revealed a similar trend only when prescribed by out-of-state practitioners.

#### 3.2.2. Opioid Type and Pharmacy Type

Analyzing the influence of pharmacy type, Figure 4 underscores the augmented dispensation of fentanyl in the high-risk areas by both chain and retail pharmacies, with the role of chain pharmacies being more crucial. In contrast, for natural and semi-synthetic opioids, a reduced dispensation was observed in the high-risk areas for both pharmacy types, where chain pharmacies uniquely exhibited a declining trend, indicating a receding dependency on them for such drugs.

#### 3.2.3. Pharmacy Type and Prescriber Location

Incorporating prescriber location, Figure 5 reveals an interesting dynamic: chain pharmacies were notably accommodating more prescriptions from out-of-state doctors, while retail pharmacies maintained a more steady approach. Notably, there was a discernible decline in prescriptions from in-state doctors, which was primarily seen in chain pharmacies.

#### 3.2.4. Multifactorial Analysis

Figure 6 depicts how the percentage difference in per capita prescriptions between the low- and high-risk counties changed over time (calculated on a monthly basis) with respect to the drug type and prescriber location. The results reaffirmed the pronounced preference of the high-risk patients for out-of-state doctors and the prominent role of chain pharmacies in filling such prescriptions, especially for fentanyl, displaying an escalating trend over time. In contrast, retail pharmacies maintained a stable approach, primarily with fentanyl prescriptions from out-of-state doctors. Additionally, there was a marginal inclination for acquiring fentanyl from in-state doctors irrespective of pharmacy type; however, this upward trajectory was significantly noticeable in retail pharmacies. Meanwhile, for natural and semi-synthetic opioids prescribed by in-state doctors, a decreasing trend was more visible, notably in chain pharmacies.

Our analysis demonstrated that the high-risk counties had a growing predilection for fentanyl, notably from out-of-state prescribers, and chain pharmacies were increasingly central to this pattern. While retail pharmacies exhibited a more balanced and consistent behavior across different opioids and prescriber locations, the overall multifactorial analysis elucidated nuanced trends and preferences in opioid prescriptions in the high-risk counties.

#### 3.2.5. Impact of Health Infrastructure Accessibility

Throughout our study, we extensively examined the legal prescription data and identified distinctions between the the high- and low-risk counties in South Carolina, with the high-risk counties characterized by below-average access to medications for opioid use disorder (OUD) and above-average overdose rates.

A pivotal finding was the prevalent tendency of individuals in the high-risk counties to consult with out-of-state doctors, which was likely attributed to the insufficient and fragile health infrastructure in these regions. This was corroborated by the lower availability of medications for OUD and a lower MD density, with the state average being 161.45 doctors per 100,000 residents. In the high-risk areas, this number dwindled to 137.36, while in the low-risk areas, it increased to 167.31. The associated *p*-value of 0.0440 signified a statistically significant disparity in the MD density between the high- and low-risk areas. In our study, the MD density was calculated specifically at the county level. This means we determined the number of MDs per 100,000 population within each county. This county-level analysis allowed us to assess the variations in MD availability and distribution across different geographic areas within our study region.

Additionally, our analysis results suggest doctor shopping may be more prevalent in the high-risk counties, predominantly involving out-of-state doctors. On average, there were 80.89 instances of per capita doctor shopping out-of-state, but in the high-risk counties, this figure significantly surged to 90.92, while in the low-risk counties, it was approximately 78.45, with a *p*-value of 0.006, confirming a statistically significant difference. Lastly, it is crucial to acknowledge that not all overdoses were attributed to prescribed medications; a substantial part was due to illegal supplies, which was a facet not considered in this analysis.

## 4. Discussion

Our research delved into the multifaceted nature of South Carolina’s opioid crisis using data that spanned from 2017 to 2021. Consistent with prior studies, we observed a decline in opioid prescriptions statewide; however, there was a notable increase in opioid-related overdose deaths [52]. Of particular concern was the rise in fatalities associated with fentanyl, which is a potent synthetic opioid. While fentanyl was prescribed less frequently compared with natural and semi-synthetic opioids, like hydrocodone, oxycodone, and methadone, its associated fatalities rose significantly. This surge in deaths linked to synthetic opioids, primarily fentanyl and its analogs, alongside methamphetamine, marks what has been termed the "fourth wave" of the opioid crisis [53]. Our study revealed that between 2017 and 2021, opioid prescriptions decreased from 64,223 to 41,214 per 100,000 residents, while fentanyl-related overdose deaths increased by 312%. The high-risk counties exhibited notably higher rates of fentanyl prescriptions and relied more on out-of-state doctors. Additionally, they demonstrated elevated instances of doctor shopping, fewer medical doctors per capita, and limited access to MOUDs. To effectively address the opioid crisis, we advocate for enhanced local healthcare infrastructure, expanded treatment access, tighter management of out-of-state prescriptions, and the meticulous tracking of prescription patterns. Tailored local strategies are vital for mitigating the opioid epidemic in these communities.

Examining the situation at the county level introduced additional layers of complexity, confirming earlier observations [54]. The high-risk counties were distinctive in the manner in which opioids were sourced and consumed. A notable difference was the increase in prescriptions that originated from out-of-state physicians. This increasing trend suggests that individuals were expanding their networks beyond state lines to obtain opioids. The weaker health systems and a shortage of doctors in the high-risk counties likely played a role in this reliance on out-of-state prescribers. There is an urgent need to strengthen the healthcare systems in these high-risk counties, making healthcare services more available and reducing the reliance on external prescribers.

Our findings also suggest the possibility of unauthorized activities, like doctor shopping. This behavior calls for actions that extend beyond the boundaries of South Carolina. Supported by our previous studies, these out-of-state resources often exhibit more lenient regulations, introducing an additional layer of risk [29,30]. Furthermore, the presence or absence of MOUDs is key to understanding these supply trends. The restricted access to MOUDs in high-risk counties, even when people are ready to seek treatment, underscores the imbalance between supply and demand in treatment options, potentially escalating the opioid issue [26].

Regarding pharmacies, the prominent role of chain pharmacies in filling fentanyl prescriptions from out-of-state sources and the consistent rate at retail pharmacies underscore the significance of pharmacy chains in these regions. As this trend is only apparent for fentanyl and not for natural and semi-synthetic drugs, it necessitates closer scrutiny and enhanced regulations on synthetic opioids in high-risk areas. Strengthening local regulations could be a quicker solution compared with expanding the health infrastructure. This trend contradicts previous findings suggesting a decrease in opioid distribution when an independent pharmacy becomes part of a chain [55].

### 4.1. Policy Recommendations

To effectively combat the opioid crisis, it is crucial to enhance the local healthcare infrastructure by increasing funding and offering financial incentives to attract healthcare professionals to underserved areas. This can be achieved through the allocation of more state and federal funding to high-risk counties to build and sustain healthcare facilities, and implementing loan forgiveness programs to incentivize healthcare providers to work in these regions. Additionally, expanding access to MOUDs is vital. Initiatives should be developed to train and certify more healthcare providers in high-risk counties to prescribe and manage MOUDs, and telehealth services should be utilized to provide MOUDs and other addiction treatments, particularly in remote and underserved areas. Furthermore, tightening the control of out-of-state prescriptions through robust data-sharing agreements between states and strengthening PDMPs will help to track prescriptions across state lines and flag suspicious activities, such as doctor shopping. Stricter regulation of synthetic opioids, especially fentanyl, is also necessary. This can be done by increasing the regulatory scrutiny on pharmacies, particularly chain pharmacies, to ensure compliance with opioid dispensing laws, and developing policies specifically targeting synthetic opioids, including stricter prescription guidelines and monitoring. Finally, community-based interventions are essential. Launching educational campaigns to raise awareness about the risks of opioid misuse and the availability of treatment options, and establishing community support services, including counseling and rehabilitation programs tailored to the needs of high-risk communities, will help to address the crisis at the local level.

### 4.2. Future Research Directions

Our study had its limitations. The absence of patient-level data prevented us from drawing firm conclusions regarding the occurrence and specifics of OUD. Additionally, the lack of data on illegal drug supply hampered our comprehensive understanding of the opioid issue, given this study focused solely on legal prescriptions. In future studies, it is crucial to further investigate several key areas to enhance our understanding and response to the opioid crisis. Patient-level data analysis should be prioritized to accurately track OUD incidence and treatment outcomes, especially in high-risk counties [30]. Understanding illegal drug supply dynamics, including the distribution and availability of synthetic opioids, requires collaboration with law enforcement agencies to effectively analyze seizure data. Evaluating the effectiveness of interventions, such as increased healthcare funding and stricter prescription controls, through longitudinal studies is essential to establish best practices. Furthermore, exploring socioeconomic and demographic factors influencing opioid misuse rates, particularly poverty and education levels, can guide tailored intervention strategies. By pursuing these research avenues, stakeholders can develop a more comprehensive approach to combating the opioid crisis, ultimately reducing misuse and overdose deaths.

## 5. Conclusions

In summary, our study revealed a complex landscape of opioid use in South Carolina from 2017 to 2021. Despite a statewide decrease in opioid prescriptions, overdose fatalities—particularly from fentanyl—surged. High-risk counties, which were distinguished by a limited access to MOUDs, exhibited unique trends: a greater percentage of fentanyl prescriptions and a significant reliance on out-of-state doctors for opioid prescriptions. This raises concerns about the efficacy of state-level interventions alone and suggests the need for broader, possibly national, strategies to address the crisis, focusing not just on regulating the supply but also on improving MOUD accessibility.

## Figures and Tables

**Figure 1 healthcare-12-01268-f001:**
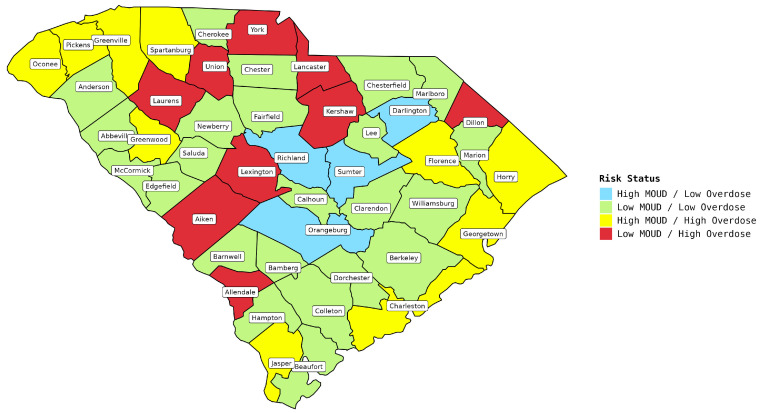
Risk statuses of counties in South Carolina.

**Figure 2 healthcare-12-01268-f002:**
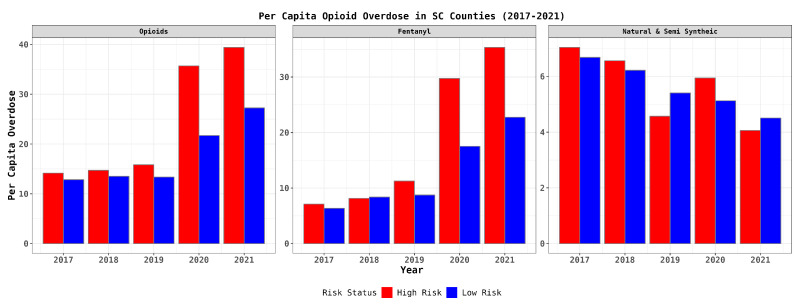
Temporal trend of county-level overdose by underlying cause of death during years 2017 to 2021—state of South Carolina.

**Figure 3 healthcare-12-01268-f003:**
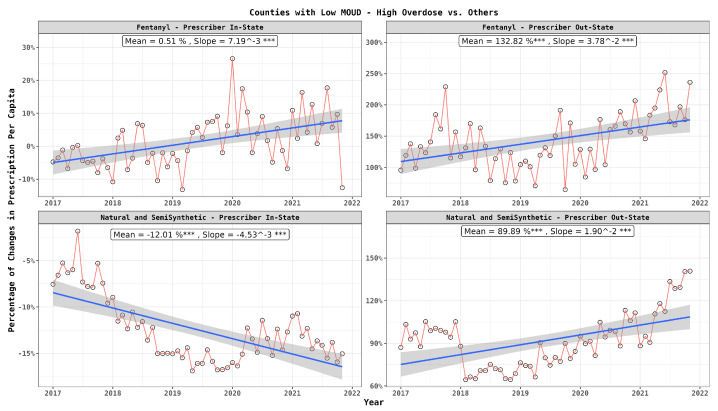
Monthly Per capita prescription differences between patients in high-risk and low-risk counties: drug type and prescriber location. In the graph, statistical significance is indicated next to the slope values as follows: ‘***’ for highly significant differences (p≤0.001).

**Figure 4 healthcare-12-01268-f004:**
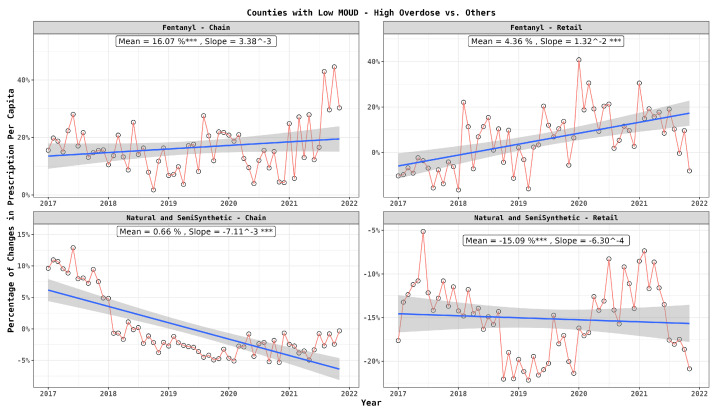
Monthly per capita prescription differences between patients in high-risk and low-risk counties: drug type and dispenser type. In the graph, statistical significance is indicated next to the slope values as follows: ‘***’ for highly significant differences (p≤0.001).

**Figure 5 healthcare-12-01268-f005:**
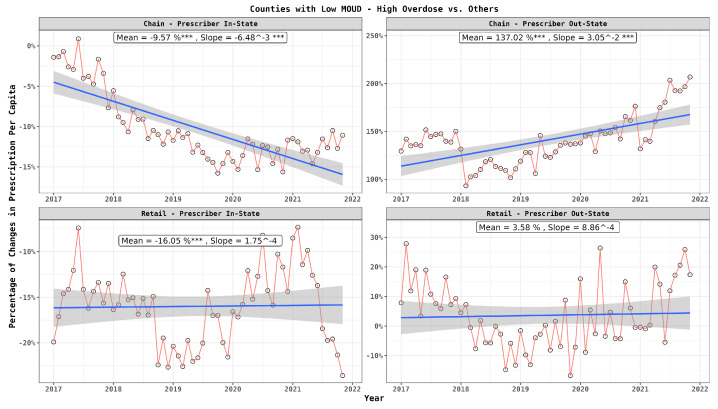
Monthly per capita prescription differences between patients in high-risk and low-risk counties: dispenser type and prescriber location. In the graph, statistical significance is indicated next to the slope values as follows: ‘***’ for highly significant differences (p≤0.001).

**Figure 6 healthcare-12-01268-f006:**
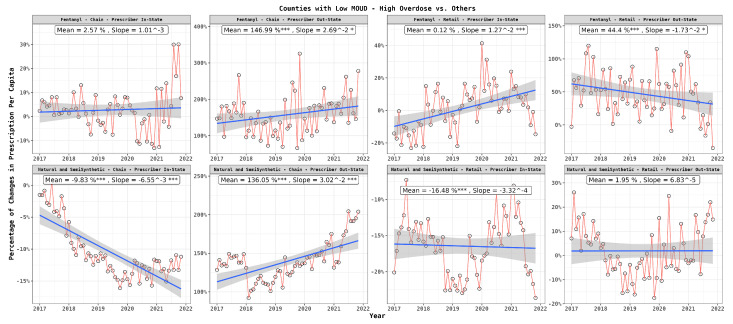
Monthly per capita prescription differences between patients in high-risk and low-risk counties: drug type, dispenser type, and prescriber location. In the graph, statistical significance is indicated next to the slope values as follows: ‘***’ for highly significant differences (p≤0.001), ‘*’ for significant differences (p≤0.05).

**Table 1 healthcare-12-01268-t001:** Opioid prescription rates per 100,000 residents during years 2017 to 2021—state of South Carolina.

Drug Class	Year					Change (%)
	2017	2018	2019	2020	2021	
Opioids	64,223	54,778	50,514	47,686	41,214	−35.83%
Natural and semi-synthetic	58,294	49,437	45,467	42,957	37,310	−35.99%
Fentanyl	1490	1206	1097	970	772	−48.19%

**Table 2 healthcare-12-01268-t002:** Opioid overdose deaths during years 2017 to 2021—state of South Carolina.

Drug Class	Year					Change (%)
	2017	2018	2019	2020	2021	
Total drug overdoses	1001	1103	1131	1734	2168	117%
Prescription drugs	782	863	923	1463	1853	137%
Opioids	748	816	876	1400	1733	131%
Natural and semi-synthetic	315	332	349	418	373	18%
Fentanyl	362	460	537	1100	1494	312%

**Table 3 healthcare-12-01268-t003:** Comparison of per capita opioid prescription between high-risk and low-risk counties.

Characteristic	Mean (SD)			*p*-Value
	All Counties	High-Risk Counties	Low-Risk Counties	
Drug Type				
All opioids	57,625 (17,154)	58,974 (16,736)	57,250 (17,295)	0.5243
Natural and semi-synthetic	52,439 (1550)	51,060 (13,906)	52,774 (16,426)	0.4769
Fentanyl	1295 (539)	1465 (603)	1248 (512)	0.0233
Prescriber Location				
In-state doctors	52,604 (17,500)	51,427 (20,002)	52,932 (16,787)	0.6282
Out-of-state doctors	5020 (5127)	7547 (6698)	4319 (4367)	0.0020
Dispenser Type				
Retail	16,356 (9372)	17,915 (9823)	15,923 (9224)	0.2029
Chain	32,796 (132,389)	32,713 (9559)	32,819 (14,114)	0.9505

## Data Availability

Data supporting the results reported in this article can be found through various public sources. Information on overdose deaths can be accessed at https://justplainkillers.com/data/. Data regarding Opioid Treatment Facilities are available at the following links: for OTP data, visit https://dpt2.samhsa.gov/treatment/; for buprenorphine treatment practitioners, refer to https://www.samhsa.gov/medication-assisted-treatment/find-treatment/treatment-practitioner-locator; and for naltrexone treatment information, access https://www.vivitrol.com/, (accessed on 18 June 2024). However, data from the South Carolina Prescription Tracking System known as SCRIPTS is not available due to ethical and privacy concerns.

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
