# Peer review of "Exploring Opioid Prescription Patterns and Overdose Rates in South Carolina (2017–2021): Insights into Rising Deaths in High-Risk Areas"

_healthcare, 2024, doi:10.3390/healthcare12131268_

Round 1

Reviewer 1 Report

Comments and Suggestions for Authors

Author Response

Dear reviewer,

Please find the attached document for our response letter. Your comments and suggestions will greatly contribute to the refinement of our research. Should you have any questions or require additional information, please do not hesitate to contact us.

Thank you very much for your time and consideration. We look forward to receiving your feedback.

Bests,

Reviewer 2 Report

Comments and Suggestions for Authors

I read with interest the paper titled "Exploring Opioid Prescription Patterns and Overdose Rates in South Carolina (2017-2021): Insights into Rising Deaths High-Risk Areas"

The paper is well written and I just want to add some minor comments: 

1 In the Abstract, please include a brief mention of statistical methods and a more detailed conclusion of what authors found. 

2 - Are the overdoses described as intentional or accidental? Or are them just classified as overdose?

3 - In this study, authors analyzed three specific medications for OUDs: methadone,

buprenorphine, and naltrexone. Why they choose those ones? In a recent study of Gustafsson et al. the overdose as described with a OR of 19.8 for Fentanyl, and 7.6 for Morphine. Curiously, the OR for oxycodone was 3.1, meaning that the risk of overdose is not that high for oxycodone, compared with other opioids. The choose of those three drugs should be clarified and the results should be discussed, comparing with other studies (as suggestion read:

a) - Chiappini et al. (2022). Pharmacovigilance signals of the opioid epidemic over 10 years: data mining methods in the analysis of pharmacovigilance datasets collecting adverse drug reactions (ADRs) reported to EudraVigilance (EV) and the FDA Adverse Event Reporting System (FAERS). Pharmaceuticals, 15(6), 675.

b) - Schifano, F., Chiappini, S., Corkery, J. M., & Guirguis, A. (2019). Assessing the 2004–2018 fentanyl misusing issues reported to an international range of adverse reporting systems. Frontiers in Pharmacology, 10, 428774.

c) - Gustafsson, M., Matos, C., Joaquim, J., Scholl, J., & van Hunsel, F. (2023). Adverse drug reactions to opioids: a study in a national pharmacovigilance database. Drug Safety, 46(11), 1133-1148.

4 - Strengthen the discussion with specific policy recommendations or future research directions.

Author Response

(The authors gave the same response as above.)

Reviewer 3 Report

Comments and Suggestions for Authors

The study is a good effort to screen and understand the utilization and effects of opioid use in South Carolina, and its counties over four years. The abstract covers the study and the issue under consideration. The introduction mentions the importance and seriousness of the study. Specifically, the concept of "Doctors Shopping" and its county-wise distribution is fundamental. Almost in every quarter of the globe, we observe the same situation. 

to control the use and overdose of the opioid, it is imperative to cover the study in all aspects. A matter of focus was the out-of-state prescriptions in low-risk areas. 

I think such studies help manage the misuse, abuse, and overdose of controlled substances. At the same time, I would suggest having current data on the issue.  

The references are majorly old, seems that the write-up was done years ago. Some more references should be added in the past five years or so.

please check the reference styles from 28 to 36, ignore if it is correct. Please see if 28-31 are not depicting self-citation.

Comments on the Quality of English Language

Some composition corrections are required, but overall it is fine. 

Author Response

(The authors gave the same response as above.)

Reviewer 4 Report

Comments and Suggestions for Authors

A very well written manuscript on a much needed topic. The authors aim to examine the legal supply side of the opioid crisis using county-level data for South Carolina from 2017 to 2021. The authors have employed a very rigorous data collection method and statistical analysis.

While it is termed "doctor shopping", maybe time to rename it as "Provider shopping" as the prescribers are not exclusively physicians. Throughout the manuscript, the authors refer to prescriptions from doctors, but would be better to term it as "providers".

Do authors have data on who the prescribers are - MD vs DO vs mid levels like nurse practitioners/physician assistants ? This would be a very important piece of information, as we are seeing decrease in prescription rate by MDs, but higher by mid levels. If this data is available, it should be included.

The authors describe chain pharmacy and retail pharmacy. What about hospital affilitated pharmacies, can those be differentiated ? Is data available on ER visits for patients and a prescription for opioid after that ?

Page 9: the authors state ". The associated P-value of 0.0440 signifies a statistically significant 314 disparity in MD density between high and low-risk areas". How is the MD density calculated ? This is not described. And again, MDs are not the only prescribers.

Retrospective data base limitations need to be described more in the limitations sections. The inability to link patient level data between all the databases is a big limitation. A multivariate analysis would be the best way to analyze big data, but with unavailability of patient level data, this is not possible here and needs to mentioned under limitations.

Author Response

(The authors gave the same response as above.)
